# The Effects of Sheet Thickness and Excitation Frequency on Hysteresis Loops of Non-Oriented Electrical Steel

**DOI:** 10.3390/s22207873

**Published:** 2022-10-17

**Authors:** Krzysztof Roman Chwastek

**Affiliations:** Faculty of Electrical Engineering, Częstochowa University of Technology, Al. Armii Krajowej 17, 42-201 Częstochowa, Poland; krzysztof.chwastek@pcz.pl

**Keywords:** ferromagnetic hysteresis, dynamic extension, modeling

## Abstract

The paper focuses on modeling the rate dependence of hysteresis loops in conductive magnetic materials. The concept, which was advanced about fifty years ago by Chua, is discussed. It is shown that the viscous-type equation considered by Zirka and co-workers belongs to the class of Chua-type models. The dynamic effects are described with a simple fractional power law. The value of the exponent in the above-mentioned power law may be assessed on the basis of measurements of coercive field strength at different excitation frequencies. To verify the usefulness of the approach, the measurements of hysteresis loops were carried out at several excitation frequencies under standardized conditions for two grades of non-oriented electrical steel. The modeled curves are in a good correspondence with the measured ones. The considered model uses fewer parameters than approaches based on three-term loss separation schemes.

## 1. Introduction

Hysteresis loops are an important feature not only for magnetic materials, but also for diverse smart materials used in sensors, nondestructive testing and structural health monitoring [1,2,3,4,5]. The present paper focuses, however, on hysteresis modeling for “classical” ferromagnetic materials, namely non-oriented electrical steel. Special attention is paid to the effect of rate-dependence, i.e., the change in loop shape upon the increase of excitation frequency. In conductive materials, the effect is due to eddy currents generated within the sample [6].

### 1.1. Basic Information on Non-Oriented Electrical Steels

Non-oriented (NO) or alternator electrical steels are the most commonly used soft magnetic materials (around 80% of total volume [7], market value estimated at USD 12.57 billion in 2020, with a Compound Annual Growth Rate of around 5.19%, which is expected to reach USD 18.24 billion by 2028 [8]). They are typically applied as core materials in rotating electrical machines, where isotropic properties are one of the crucial requirements [9]. The range of currently manufactured NO sheets covers several grades, differing in thickness and the percentage content of silicon and aluminum (usually 1.0–3.7%wt. Si, 0.2–0.8%wt. Al).

Despite electrical devices and machines usually featuring high efficiency, the total energy loss in electrical power engineering systems may attain substantial values, which leads to quick replenishment of primary energy sources and high costs of electrical energy. The most important material used in electrical devices is electrical steel, which is used for setting up magnetic circuits. During the magnetization process, a portion of the energy is transformed into heat. This part is commonly referred to as iron losses.

According to Moses [10], losses in soft magnetic materials constitute around 5% of the total energy produced worldwide. Therefore, it is important to carry out research aimed at better understanding of hysteresis phenomenon and energy dissipation processes in magnetic circuits, since this may lead to a lowering of energy consumption and the promotion of pro-ecological solutions. These issues have been reflected in normative acts of the European Commission [11].

Table 1 schematically depicts the applications of electrical steels.

### 1.2. Basic Information on the Examined Steel Samples and the Measurement Setup

The subjects of the study presented in the present paper are two grades of NO steel exhibiting around 3.2%wt. Si. They differ in thickness (0.35 and 0.65 mm) and magnetic properties, as indicated in Table 2, which presents their basic catalogue data.

Figure 1 presents some of the measured hysteresis loops for the considered grades at $B_{m}=1.5$ T and mains frequency (50 Hz in Poland). The measurements were carried out using a Single Sheet Tester device in accordance with the requirements of the IEC 60404-3 standard. The system MAG-RJJ-2.0 [12] makes it possible to determine the properties of soft magnetic materials with the following fundamental parameters:Frequency range 5–400 Hz, with a resolution Δ*f* = 0.01 Hz and setting accuracy 0.2%;Magnetic induction (or polarization) range 0.05–2.0 T, with a resolution Δ*B* = 0.01 T and setting accuracy 0.1%;Admissible deviation of the secondary voltage shape factor from the pure sine signal does not exceed 0.1%. The accuracy of measurements meets the requirements of IEC and DIN (German) standards in the range prescribed by the standards. The guaranteed expanded uncertainty of type B for loss measurements is below 1.5% (for the 0.95 confidence level).

The basic components of the MAG-RJJ-2.0 system are presented in a diagram in Ref. [13].

Generally speaking, the IEC 60404-3 standard prescribes that the average sine polarization waveform in the sample be preserved [12,14]. The dimensions of the samples are 500 × 500 mm. In the figure, a characteristic distortion of the hysteresis loop (the “gooseneck” effect) for the thicker sample is clearly visible.

### 1.3. Problem Statement

#### 1.3.1. Some Useful Quasi-Static Hysteresis Models and the Mysterious Negative Susceptibility Region Observed for Dynamic Loops

From a causal perspective, the constitutive relationship B=μ(H+M) (Sommerfeld’s notation) or B=μH+J (Kennelly’s notation) is interpreted as follows: the application of field strength H results in an increase of the value of magnetic moments per unit volume (magnetization) aligned with the field strength vector; finally, the resulting material response is reflected in an increase of flux density (magnetic induction B). This corresponds to the so-called forward model, cf. [15]. In this case, it is natural to speak of hysteresis in terms of a lagged response between H and B. On the other hand, the conditions for material characterization in accordance with international standards assume a specific waveform shape of the system output quantity (for soft magnetic materials polarization is practically equal to induction, the difference in values usually does not exceed a fraction of a percent), and thus it requires the inversion of the computation chain, cf. [15]. Therefore, a conclusion may be drawn that hysteresis models used to describe the measurement results should have magnetic induction as their input. Examples of such models may be:The GRUCAD description [16], thoroughly examined in the context of the present paper in Ref. [17];The Harrison model [18];The simple mathematical tool T(x) model applied in reversed fashion [19];A phenomenological description similar in spirit to the above-mentioned T(x) model, but based on the arctangent function [20], in which an implicit dependence of some parameters on frequency is implemented [21];The Tellinen model [22] and its recent modifications [17,23]; andWith some reservations, the inverse Jiles–Atherton approach [24,25,26,27] (the problems related to the uncritical use of this description are briefly outlined in Refs. [28,29]).

At this point, it can be remarked that there is some controversy regarding the straightforward use of dynamic extension to the Preisach model, as considered, e.g., in Ref. [30]. The Preisach model has field strength as its natural input signal; thus, in order to adapt it to true measurement conditions, it might be necessary to determine the so-called inverse distribution function [31] first.

Figure 2 depicts measured minor loops ($B_{m}=0.5$ T) for the considered steel grades at several excitation frequencies. As the excitation frequency increases, the hysteresis loop branches begin to lose their monotonous character, and the loops begin to resemble ellipses (this explains the success of Zakrzewski’s elliptical loop description and a relatively simple way to carry out loss separation for this model [32]). It is noticeable that there are regions exhibiting negative susceptibility, which may be troublesome for the analysis, especially if one is interested in incorporating hysteresis codes into finite element modeling routines.

The aim of the present paper is to verify whether there exists a simple method to take into account the peculiar shape of dynamic hysteresis loops into account, no matter that this issue is considered to be “too ambitious” by the author of a well-known monograph [33], page 391.

#### 1.3.2. To Separate or Not to Separate? That Is the Question

An issue closely related to the problem tackled in the present paper is the description of energy losses in soft magnetic materials under increased excitation frequency, since the loop area corresponds to energy converted into heat during magnetization [34]. Figure 3 schematically depicts some possible approaches. Figure 3a depicts the quasi-static hysteresis loop, obtained for slow excitation frequency. Figure 3b depicts the hysteresis loop under increased excitation frequency, where it can be evidently seen that the loop is wider (in particular the coercive field strength increases). This scenario assumes that there is no physical distinction between the processes under DC conditions and under AC conditions; thus, “the distinction between hysteresis loss and AC or eddy-current loss is imaginary” [35], page 449. In a similar spirit, Becker wrote in 1963: “… the concept that hysteresis loss is something that can be added to eddy-current loss implies that it has a different independent mechanism, operating at all times and independent of frequency. However, as we have been trying to emphasize so far, there is only one mechanism of losses in these materials, namely resistive losses due to eddy currents associated with moving walls. Since the number of moving walls changes with frequency, it is difficult to see what meaning should be attached to the separation of a “hysteresis” loss involving few moving boundaries from another loss involving more, or even, if the collapsing wall region is reached, an entirely different configuration” [36].

To derive an extension of hysteresis model to dynamic conditions, it might be necessary to identify a single parameter of the hysteresis model that controls coercive field strength, and subsequently to apply a functional dependence on frequency to this parameter [37,38]. A more sophisticated approach, behaving in accordance with Figure 3b, may rely on the introduction of a rate-dependent term in the so-called “effective field”, accounting for coupling effects [39,40]. It is remarkable that this approach is able to describe the “gooseneck” effect shown in Figure 1, cf. [39]. On the other hand, in the course of research, it was found that it is valid over a more limited frequency range in comparison to the approach in which the hysteresis and viscous effects are considered independently (scenario depicted in Figure 3c), i.e., a two-term description, cf. [41].

A two-term description of energy loss in ferromagnets has a solid theoretical background, namely the Poynting theorem, in which only two components appear [42,43,44]. It should be remarked that recently some independent analyses have confirmed the validity of the decomposition of total energy losses into just two components, cf. [45,46]. Taking into account the flexibility offered by this approach, in the present paper, it is chosen for computation purposes.

Finally, Figure 3d depicts the description based on the so-called three-term separation, which nowadays is commonly attributed to Bertotti [33]. In fact, the framework is much older and can be traced back to the concepts found in Legg’s [47] or Stewart’s [48] papers. According to Bertotti’s approach, it is possible to distinguish energy losses from macro eddy currents flowing in whole bulk materials (determined by the sample geometry) from energy losses related to micro eddy currents around moving domain walls; this leads to different contributions in the total energy balance. Energy terms have their counterparts in field strength components. Despite this approach having been found to be successful in the description of dissipation phenomena in soft magnetic materials by some researchers [13,26,27,49,50], in the present paper it is excluded from analysis for just two reasons:According to Occam’s razor principle, the description should be made as simple as possible, and the three-term description has some limitations due to the presence of the “classical” loss term, which in its original form does not account for corrections due to eddy current shielding; therefore, it should be considered as approximate only. Already in 1963 Becker warned against an uncritical use of three-term separation schemes: “…if the losses are being calculated on a model that is not correct, it does not appear fruitful to regard the discrepancy as a third independent component of the total loss” [36].The origin of excess loss as related to eddy currents has been put into question by Mayergoyz and Serpico, who regarded the third term as being related rather to the intrinsic nonlinear dynamics underlying bistable/multistable behavior associated with hysteresis [51].

Taking into account the interpretational problems related to three-term loss separation, in the subsequent part of the manuscript, all viscosity effects are lumped together, and thus the scenario depicted in Figure 3c is considered.

#### 1.3.3. How to Extend a Hysteresis Model to Dynamic Conditions

The extension may be carried out seamlessly by introducing additional offsets of loop branches dependent on B and dB/dt. This concept is applicable to any reliable hysteresis model. The approach was studied by Chua and Stromsoe in the context of lumped circuit models for iron-core nonlinear inductors [52]. Chua’s concept was scrutinized by Saito et al., who carried out 3D analysis of nonlinear magnetic circuits using reluctance networks [53]. Some researchers [54,55,56,57,58] have considered dynamic extensions of the existing quasi-static hysteresis models, which may be traced back to the general form of Chua’s description:(1)$$\frac{dy}{dt}=g\left[x(t)-f(y(t))\right]$$
in which f(⋅) is the dissipation function and g(⋅) is the restoring function, responsible for irreversible and reversible effects, respectively [52]. In the subsequent part of the manuscript, some of those extensions are discussed in detail.

Raulet et al. solved the diffusion equation using the finite difference method for the simplified, linearized relationship:(2)$$\frac{dB}{dt}=\frac{1}{\beta }\left[H(t)-H_{\mathrm{stat}}(B)\right]$$
where $H_{\mathrm{stat}}(B)$ was computed from a quasi-static material response and β was assumed to be constant [59].

Fujiwara and Tahara examined the dependence of the equivalent field strength $H_{e}(B)=H_{\mathrm{ac}}(B)\;-\;H_{\mathrm{dc}}(B)$ for three grades of steel used in electrical engineering (grain-oriented, non-oriented 3.2%wt. Si and 6.5%wt. Si) and claimed that $H_{e}(B)$ might be given as a power law with respect to induction rate dB/dt, with the exponent value around 0.6–0.7, regardless of the material morphology [60].

Zirka et al. considered a similar relationship in Ref. [61], in which the proportionality coefficient in the power law was varied upon B:(3)$$H(t)=H_{\mathrm{stat}}(B)+{\left[\frac{1}{r(B)}\frac{dB}{dt}\right]}^{1/\nu }$$
where r(B) represented the so-called dynamic magnetic resistivity, $r(B)=K\left[1-{\left(B/J_{\mathrm{sat}}\right)}^{2}\right]$, $J_{\mathrm{sat}}$ is the saturation polarization, and the fractional exponent 1/ν accounts for model dynamics. The validity of this approach was subsequently also studied in publications co-authored by the author of the present contribution, such as [62,63].

As a general remark, it can be stated that it is possible to model hysteresis curves at increased excitation frequency by adding an additional term related to eddy current damping to the quasi-static $H_{\mathrm{stat}}(B)$ dependence. This leads to a two-term loss separation formula [43], and the issue of interdependence between hysteresis and eddy currents [46,63,64] is significantly simplified. The additional term may possibly be described with a fractional power law. Such relationships have recently been the subject of considerable research [65,66,67].

## 2. Modeling

The starting point for the choice of the proper functional dependence linking dB/dt to the equivalent field strength $H_{e}(B)$ might be the analysis of the $H_{c}=H_{c}(f)$ relationship ($H_{c}$ denotes the coercive field strength).

Figure 4 depicts the measured dependencies $H_{c}=H_{c}(f)$ for the M330-35A steel grade, whereas Figure 5 depicts them for the M530-65A steel grade. The results of the straight-line fits $H_{c}=a+b\;f$ are also shown. Two conclusions may be immediately drawn from the analysis of the figures:We consider the ascending branch of the linear fit to be acceptable, in particular for higher induction levels; however, a more complicated nonlinear dependency might be more appropriate for lower induction levels, in particular for the thick sample;The slopes of the fitting lines increase with increasing induction amplitude; this fact is at odds with the constant β value assumed in Ref. [59]. The approach considered by Raulet et al. assumed a constant β value in the whole sample cross-section regardless of excitation level, which allowed the authors to implement a simple finite-difference scheme for solving the diffusion equation. The measurement results depicted in Figure 4 and Figure 5 prove that the assumption of constant β is oversimplified, since the local induction values at each “slice” of the sample (i.e., at different depths) may vary considerably; accordingly, the local β values should be updated.

For the dependencies shown in Figure 4 and Figure 5, the values of the adjusted coefficients of determination ${\bar{R}}^{2}$ exceeded 0.987, thus they were relatively high.

It can, however, be seen that a fractional power law in the form $H_{c}=H_{c0}+c\;f^{1/\nu }$ might be more appropriate, as is evident from an inspection of Figure 6, which refers to the thicker (more “nonlinear”)-grade M530-65A. The value of exponent was fixed at 1/ν=0.787. Moreover, the dependence $c=c\;(B_{m})$ was assessed. It may be estimated to be approximately $c=1.49\;B_{m}.$ Analogously, the dependence of quasi-static coercive field strength may be envisaged as a linear relationship with respect to induction $H_{c0}\;(B_{m})=19+11\;B_{m}.$ It should, however, be remarked that the relationships given above were derived from a limited number of data points, and thus they should be treated in a rather qualitative way.

A similar analysis was also carried out for the M330-35A grade. In this case, the value of exponent 1/ν was closer to unity, namely 1/ν=0.838. The proposed dependence for $c=c\;(B_{m})$ was $c=0.48B_{m},$ whereas the relationship for $H_{c0}\;(B_{m})$ was $H_{c0}\;(B_{m})=18+13.8\;B_{m}.$

It was assumed that excitation frequency *f* = 5 Hz was low enough to be neglected in the first approximation of domain wall mobility (a quantity that to a great extent affects the magnetization process [6,36,68]); thus, the measurements for this frequency can be treated as approaching the quasi-static limit, and the modeled values of coercive field strength were compared to those from measurements. The relative deviations, computed from the relationship $\delta_{H_{c0}}=100\left|H_{c0\;\mathrm{model}}-H_{c0\;\mathrm{meas}}\right|/H_{c0\;\mathrm{meas}}$, did not exceed 9.7% for the M330-35A grade and 24.7% for the M530-65A grade. The modeled values were underestimated in all considered cases (for $B_{m}=1.5$ T, and for excitation conditions approaching saturation by 9.7% for the first and by 10.6% for the second grade).

It can be noticed that the obtained values of fractional exponent 1/ν are comparable to those listed by Najgebauer [69] and slightly higher than those suggested by Fujiwara and Tahara [60].

To verify the usefulness of Chua’s extension modeled hysteresis loops for increased excitation frequencies, the measured “quasi-static” (5 Hz) magnetization curves were taken as a reference. In this way, it was possible to avoid the accumulation of additional errors resulting from inaccuracies in the quasi-static hysteresis models in the final results.

The approach consists of the following steps:It is assumed that the quasi-static B=B(H) (or more precisely H=H(B)) dependence is known for a given amplitude $B_{m}$;From the measurements of $H_{c}=H_{c}(f)$, we know the value of the fractional exponent 1/ν needed for subsequent computations;If we consider the ascending branch of the hysteresis loop under symmetric excitation, then we notice immediately that B varies from $-B_{m}$ to $B_{m}$ in accordance with $B\left(t\right)=-B_{m}\cos\omega t=-B_{m}\cos2\pi ft$, where the “local” time runs from 0 to T/2=1/(2f) . Thus, in order to reconstruct the waveform for an increased frequency, we compute dB/dt for that frequency. Substituting the known quantities into (3), there is just one unknown, namely the normalization constant K; its value is determined from a comparison of measured and modeled magnetization curves at increased frequency;The descending branch of the hysteresis loop is determined from the odd symmetry condition, H(−B)=−H(B).

It is believed that this approach also remains valid for distorted hysteresis loops (e.g., those containing higher harmonics or simply a minor loop), as long as the quasi-static hysteresis model is able to correctly capture the distortion. If the analytical dependence H=H(B) is hard to write, then it will be necessary to resort to numerical methods (numerical differentiation) to recover the waveform of dB/dt.

Figure 7 depicts the modeling results for the major loop of the M530-65A grade at *f* = 50 Hz. It can be seen that the “gooseneck” effect was not properly reproduced, the modeled loop branches are monotonous. However, the overall modeling result may be considered satisfactory in the first approximation. The absolute difference between the modeled and the measured loop areas (representing energy loss) did not exceed 6.1%.

In the next step, an attempt to model minor dynamic loops was undertaken. “Model 1” in Figure 8 denotes a scenario in which the same value of normalization constant K as for the major loop is used, whereas for “model 2” the value of K is updated. It can be noticed that in the second case, the modeled curve matches the measurements much better. Therefore, a conclusion may be drawn that it is necessary to update the value of K value on the basis of $B_{m}.$ Figure 9 depicts the modeling results for $B_{m}=0.5$ T. It can be observed that they are quite satisfactory, just like for $B_{m}=1.0$ T (model 2).

Modeling results for a major loop of the M330-35A material at mains frequency is shown in Figure 10. The model slightly underestimates the measured field strength values, but its overall accuracy is quite good.

Figure 11 depicts several hysteresis loops for a low excitation amplitude. In Figure 9, the measured curves for 5 Hz and 50 Hz may be considered to be overlapping in practice; in any case, modeling was performed for the latter frequency (not shown), and the value of K was determined. The same value was subsequently used in modeling assuming *f* = 400 Hz (model 1). There were considerable discrepancies between the modeled and the measured loops; therefore, the parameter was freed, and a new value was determined (model 2). The modeled loop better matched the measured one. A general conclusion may be drawn that K=K(B,dB/dt) should be determined individually in each considered case, since it is a function of both amplitude and induction rate; however, the determination of some useful functional dependencies is postponed to future research.

## 3. Discussion

To indirectly prove the validity of the approach presented in this paper, let us examine the dependence of total losses versus frequency using true measurement data (given explicitly for the M330-35A grade in Table 1 in Ref. [43]). The measurements shown in Figure 12 (points) were carried out for three values of induction commonly of interest to practitioners, i.e., 0.5 T, 1.0 T and 1.5 T.

Using a spreadsheet optimization routine, the value of the power law exponent was determined to be 1.39. This value slightly differs from the value 1.37, which could be inferred from the values given in Ref. [43], because in that approach, the measurement data were processed and the trends were determined from rectified formulas, so that linear regression techniques could be applied. In the present paper, the fitting was applied to raw measurement data. It was found that the fixed value of the fractional exponent is appropriate for the whole considered induction range. Thus, a conclusion may be drawn that the two-term separation formula, and consequently the addition of quasi-static and dynamic field strengths as envisaged by Chua, may lead to the correct prediction results.

To verify the usefulness of the two-term formula $P_{\mathrm{total}}=af+bf^{\alpha }$ and three-term (Bertotti’s) formula valid for NO steel, in which the exponent for the classical loss term was fixed at 2, i.e., $P_{\mathrm{total}}=cf+d+$ the averaged absolute deviations between the measured and the modeled loss values were computed using the relationship $\delta_{P\%}=100\left|P_{\mathrm{total}}^{\mathrm{meas}}-P_{\mathrm{total}}^{\mathrm{model}}\right|/P_{\mathrm{total}}^{\mathrm{meas}}$. It was found that for the two-term formula $\delta_{P\%}\;$was equal to 9.7%, whereas for Bertotti’s formula, $\delta_{P\%}=20.7\%.$ Generally, the deviations were higher for lower excitation levels in both considered cases.

A similar analysis was performed for the M530-65A grade. Figure 13 depicts the measurement points and the fit with the value of exponent in dynamic loss term fixed at 1.463. The averaged absolute deviation between the measured and the modeled data points was equal to 9.5%. The corresponding averaged value of $\delta_{P\%}\;$for the three-term formula was equal to 14.7%.

Taking into account the obtained results, it can be stated that despite the three-term separation scheme has fewer degrees of freedom (more parameters), it does not improve the modeling accuracy. In fact, the presence of the classical term in the three-term formula (which was derived for homogenously magnetized linear medium) may lead to some unexpected problems like, e.g., negative excess loss (reported in some cases for non-standard excitation in Ref. [70]) or the necessity to introduce additional functional dependencies for loss coefficients [71,72,73,74]. It is remarkable that some derived relationships are valid over a limited frequency or induction range [74]. On the other hand, the limits are often set arbitrarily by the authors themselves. From the point of underlying physics there is nothing peculiar, e.g., about the induction value $B_{m}=1\;$T and most probably the same dissipation mechanisms are valid for $B_{m}=0.95$ T or say $B_{m}=1.07$ T.

A natural candidate for comparison to the approach presented in this paper is the so-called Thin Sheet Model [23,75,76], which, apart from a fractional component, incorporates the “classical” field strength component. In this paper, we retain the simplified form for the relationship for dynamic field strength, because it is simpler and more flexible, moreover the “classical” field strength component usually does not take into account the correction due to incomplete flux penetration in the sheet, thus it is somewhat limited in usefulness. At this point the results by Broddefalk and Lindemno may be recalled [77]; these researchers found that for 0.65-mm-thick steel, the limiting frequency beyond which Bertotti’s formula fails to describe experimental data was merely 95 Hz (for $B_{m}=1.5$ T).

## 4. Conclusions

In this paper, a half-century-old concept accounting for the effect of increased excitation frequency on the shape of hysteresis loops advanced by Chua was recalled. The approach is applicable to any quasi-static hysteresis model. In order to tailor the description, it is necessary to examine the dependence of coercive field strength vs. excitation frequency. In this paper, it was shown that in the case of non-oriented steel grades, the aforementioned dependence might be described with a fractional power law with a free term. The free terms corresponding to quasi-static coercive field values and the slopes of the $H_{c}=H_{c}(f)$ curves are functions of maximum induction.

On the basis of the $H_{c}=H_{c}(f)$ dependencies and quasi-static hysteresis curves, the shapes of hysteresis loops for increased excitation frequencies can be predicted. The value of the K constant in the relationship for dynamic magnetic resistivity r(B) depends on the maximum induction and induction rate.

These results might hopefully be of interest to practitioners working on hysteresis modeling in soft magnetic materials.

## Figures and Tables

**Figure 1 sensors-22-07873-f001:**
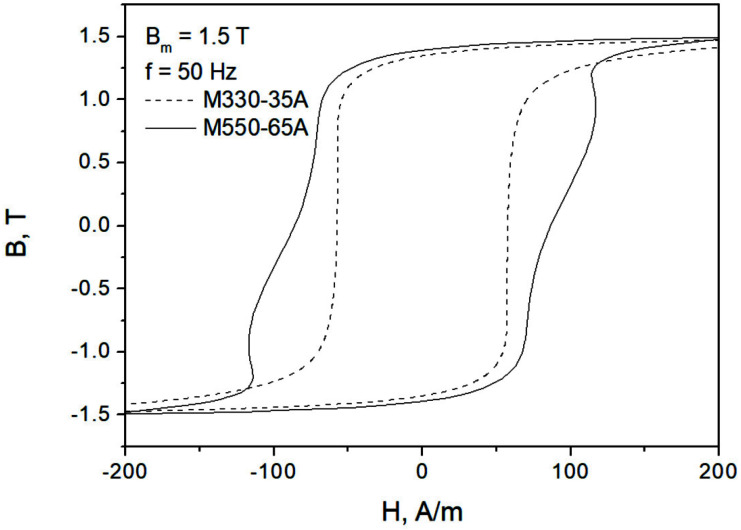
Measured hysteresis loops for two NO steel grades at $B_{m}=1.5$ T.

**Figure 2 sensors-22-07873-f002:**
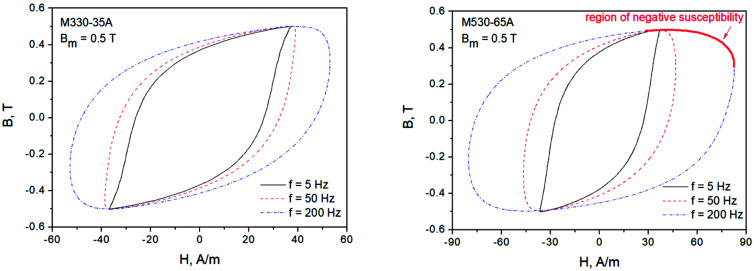
Measured hysteresis loops for two NO steel grades at $B_{m}=0.5$ T. At the increased excitation frequency, the regions with negative dynamic susceptibility are clearly visible.

**Figure 3 sensors-22-07873-f003:**
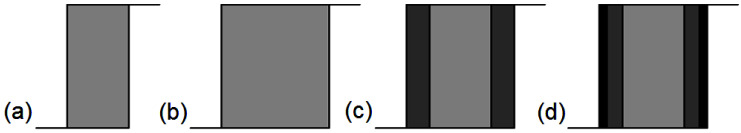
Possible approaches to addressing the change in hysteresis shape at increased excitation frequency: (**a**) quasi-static hysteresis loop, (**b**) dynamic hysteresis loop, no distinction is made as far as dissipation mechanisms are concerned, (**c**) hysteresis loss is distinguished from eddy current loss, (**d**) dynamic loss partitioning into classical and excess terms (following e.g., Bertotti’s approach [33]).

**Figure 4 sensors-22-07873-f004:**
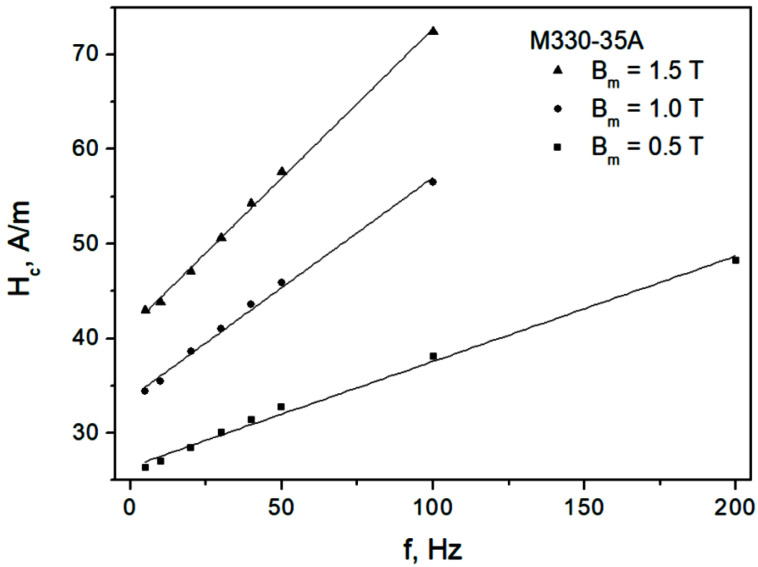
Measured $H_{c}=H_{c}(f)$ dependencies for the M330-35A grade.

**Figure 5 sensors-22-07873-f005:**
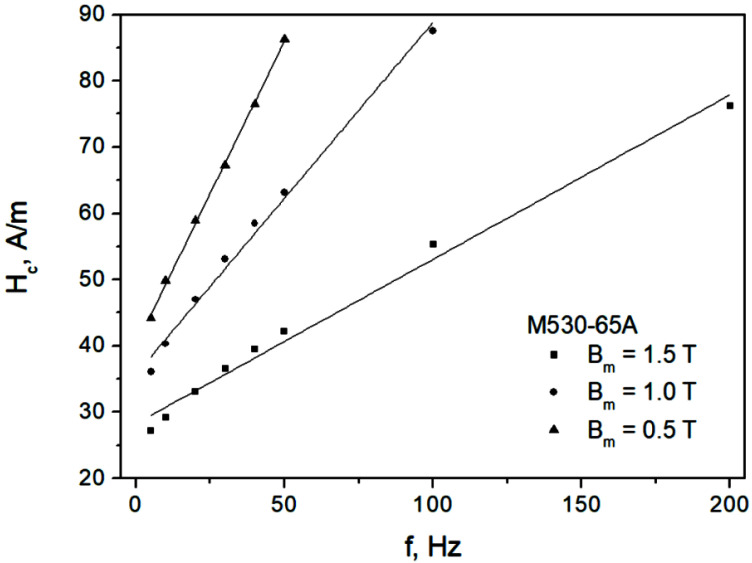
Measured $H_{c}=H_{c}(f)$ dependencies for the M530-65A grade.

**Figure 6 sensors-22-07873-f006:**
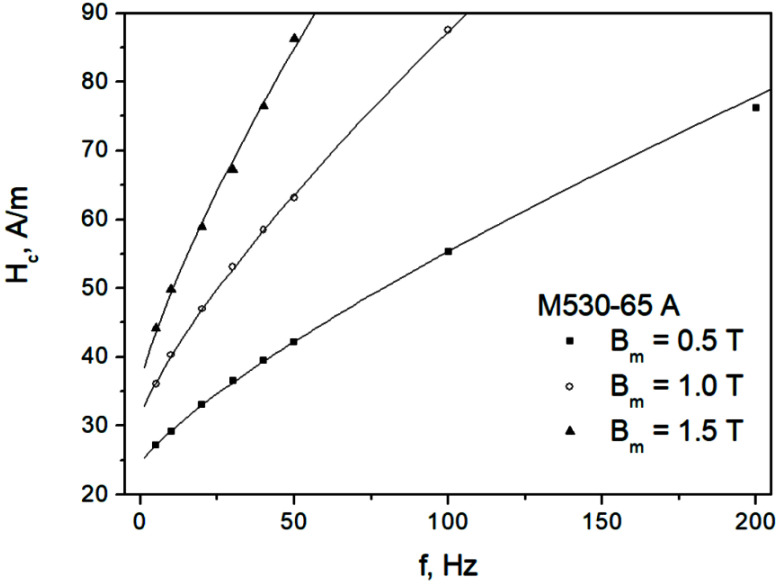
Fractional power law fit of the dependence $H_{c}=H_{c}(f)$ for the thicker steel grade.

**Figure 7 sensors-22-07873-f007:**
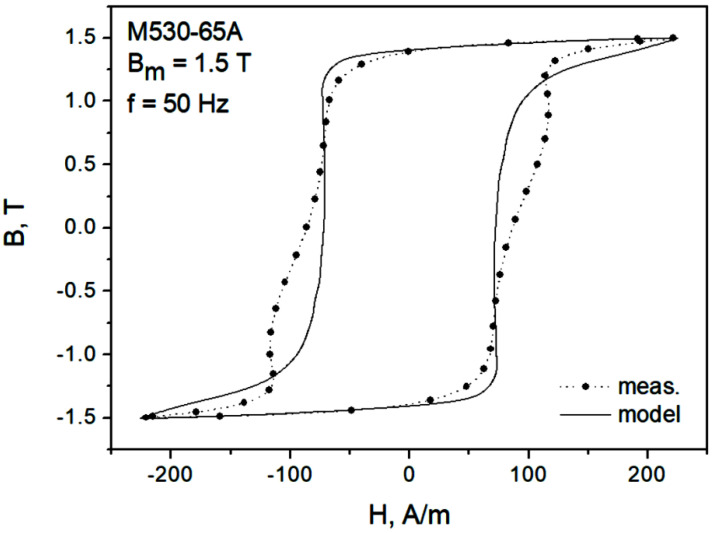
Measured and modeled hysteresis loop for Bm = 1.5 T, f = 50 Hz, M530-65A grade.

**Figure 8 sensors-22-07873-f008:**
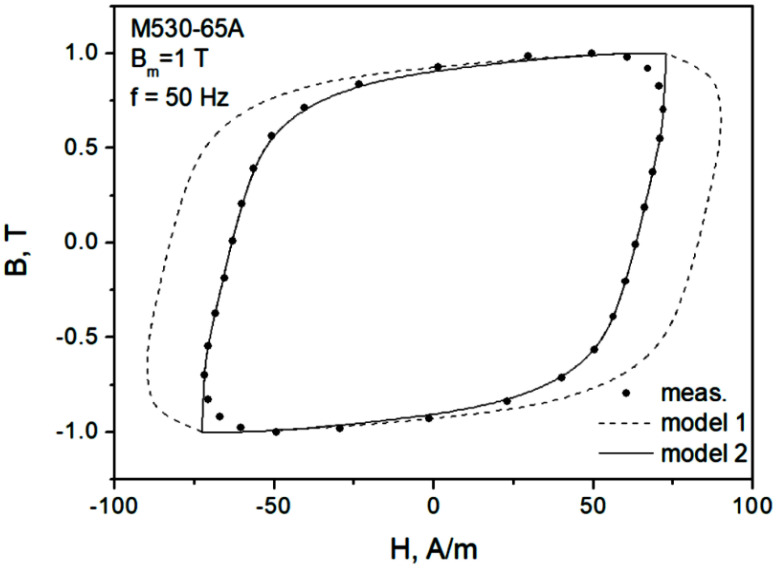
Measured and modeled hysteresis loop for Bm = 1.0 T, f = 50 Hz, M530-65A grade.

**Figure 9 sensors-22-07873-f009:**
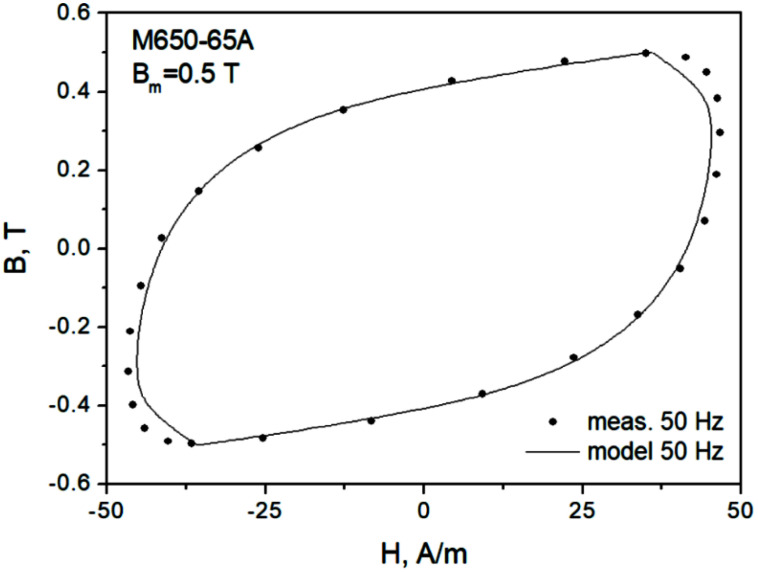
Measured and modeled hysteresis loop for Bm = 0.5 T, f = 50 Hz, M530-65A grade.

**Figure 10 sensors-22-07873-f010:**
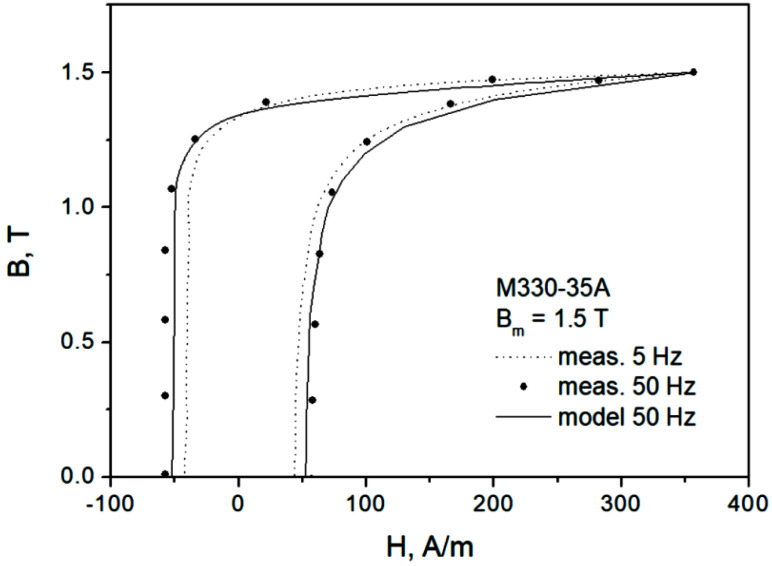
Measured and modeled hysteresis loop for Bm = 1.5 T, f = 50 Hz, M330-35A grade.

**Figure 11 sensors-22-07873-f011:**
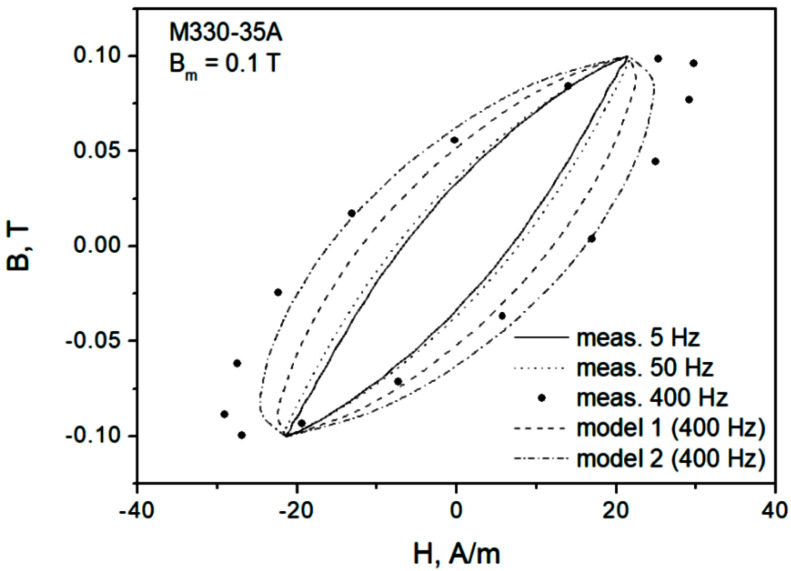
Measured and modeled hysteresis loop for Bm = 0.1 T, f = 400 Hz, M330-35A grade.

**Figure 12 sensors-22-07873-f012:**
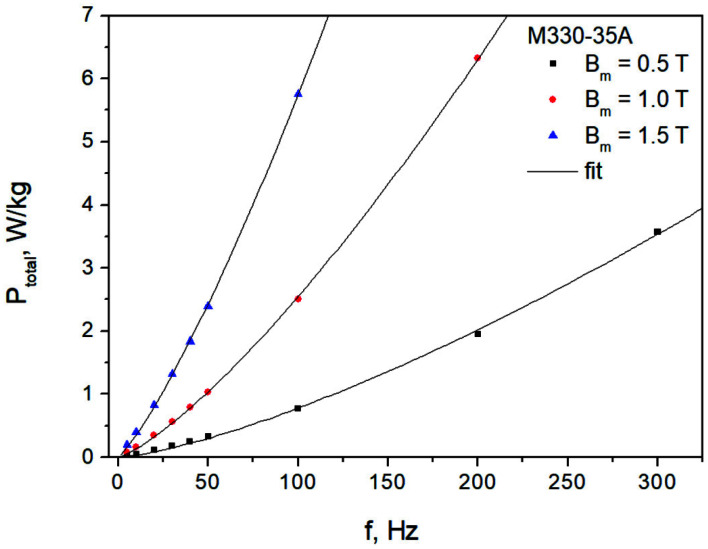
Measured and modeled P_total_ = P_total_(f) dependencies for the M330-35A steel. Fit was carried out using a fixed value of coefficient in the power law for dynamic loss.

**Figure 13 sensors-22-07873-f013:**
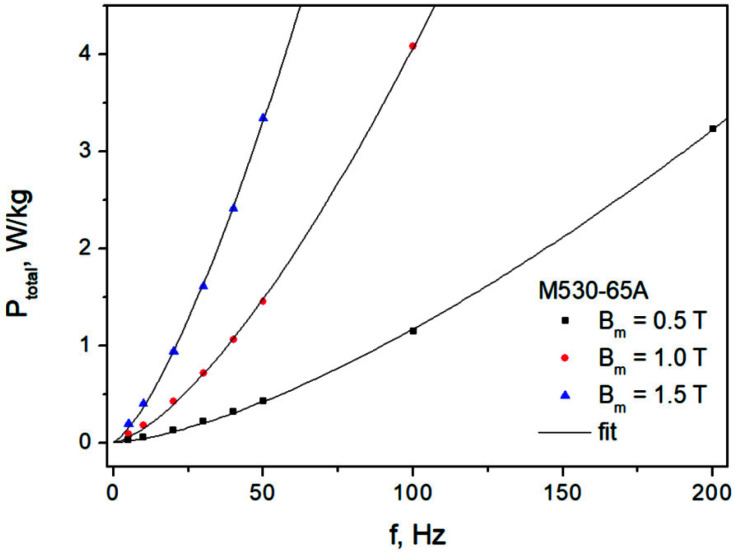
Measured and modeled P_total_ = P_total_(f) dependencies for the M530-65A steel. Fit was carried out using a fixed value of coefficient in the power law for dynamic loss.

**Table 1 sensors-22-07873-t001:** Applications of electrical steels.

**Application**	**NO Steel**	**GO Steel**
**Silicon-Less**	**Low-Silicon**	**High-Silicon**	**Conventional**	**High Permeability**
1. Small motors	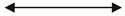			
2. AC motors, medium power	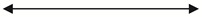			
3. Welding transformers	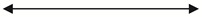		
4. Audio transformers		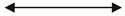		
5. Small power transformers		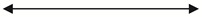	
6. Big rotating machines			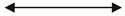	
7. Alternators and generators,medium power		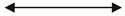		
8. Distribution transformers				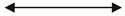
9. Power transformers				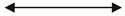

**Table 2 sensors-22-07873-t002:** Basic properties of the examined steel grades.

Thickness, mm	Grade Designation	Max. Loss Density, W/kg at f = 50 Hz		Minimal Induction, T	
		B = 1.5 T	B = 1.0 T	H = 2500 A/m	H = 5000 A/m
0.35 mm	M330-35A	3.30	1.30	1.49	1.60
0.65 mm	M530-65A	5.30	2.30	1.54	1.64

## Data Availability

Data available upon request from the author.

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
