# Peer review of "The Effects of Sheet Thickness and Excitation Frequency on Hysteresis Loops of Non-Oriented Electrical Steel"

_sensors, 2022, doi:10.3390/s22207873_

Round 1
Reviewer 1 Report
The paper is focuses on the modeling of rate-dependent hysteresis in conductive magnetic materials. The results of research show that in the case of non-oriented steel grades, the dynamic hysteresis can be described with a fractional power law with a free term. To verify the validity of the approach, the measurements of hysteresis are carried out at several excitation frequencies under standardized conditions for two grades of non-oriented electrical steel. The following modifications are suggested for the manuscript.
1. Key features of the proposed study are not clearly stated in the abstract. On the other hand, the effect of the method on performance improvement should be discussed in one sentence in the abstract.
2. Introduction section does not contain a strong motivation. In this case, it is very difficult to attract the attention of the reader.
3. Experimental details and parameters are missing. The repeatability of the experiments is therefore low.
4. Comparative analysis of the results obtained is missing. In particular, comparisons with other methods in the literature should be added.
5. In the experimental section, it is necessary to explain the setting of model parameters in detail.
6. The explanations of the experimental results are not sufficient. Especially, the curves obtained in comparative experiments are not specifically analyzed and explained.
7. Most of the references used in the paper are quite old. Please refer to the latest literature to make the research more productive, novel, and sound.
Author Response
Please find enclosed the file with detailed reply.

Reviewer 2 Report
In this manuscript, the author reviewed a half-century-old concept proposed by Chua to show the effect of increasing the excitation frequency on the shape of the hysteresis loop. It is shown that in the case of non-oriented steel grades, the dependence can be described by a fractional power law with a free term. The slope of the free term and curve corresponding to the quasi-static coercivity field value is a function of the maximum induction. The author claim that the method is applicable to any quasi-static hysteresis model. I believe that the manuscript is ready for publication, although my comments below will not materially affect the manuscript.
1. On page 6, line 209, the author mention that "the slope of the fitted line increases as the induced amplitude increases; this fact is inconsistent with the constant value assumption used in Ref. 49." It is important that more explanation should be given to explain the reason for the inconsistency.
2. Some editing errors need to be fixed, such as the missing necessary punctuation on page 8, line 254, and the stray dots in the image of Figure 2.
3. From my perspective, it is recommended that sections 1-3 be integrated, with sections 2,3 appearing in the introduction as secondary headings.
Author Response

(The authors gave the same response as above.)

Round 2
Reviewer 1 Report
The authors have revised the manuscript accordingly. The manuscript can be accepted for publication now.